# Performance-Based Pay System and Job Stress Related to Depression/Anxiety in Korea: Analysis of Korea Working Condition Survey

**DOI:** 10.3390/ijerph20054065

**Published:** 2023-02-24

**Authors:** Myeong-Hun Lim, Jin-Ha Yoon, Won-Tae Lee, Min-Seok Kim, Seong-Uk Baek, Jong-Uk Won

**Affiliations:** 1Department of Occupational and Environmental Medicine, Severance Hospital, Yonsei University College of Medicine, Seoul 03722, Republic of Korea; 2The Institute for Occupational Health, Yonsei University College of Medicine, Seoul 03722, Republic of Korea; 3Graduate School of Public Health, Yonsei University College of Medicine, Seoul 03722, Republic of Korea; 4Department of Preventive Medicine, Yonsei University College of Medicine, Seoul 03722, Republic of Korea; 5Graduate School, Yonsei University College of Medicine, Seoul 03722, Republic of Korea

**Keywords:** performance-based pay system, job stress, depression, anxiety

## Abstract

The adoption rate of performance-based pay systems has increased in recent years, and the adverse effects of systems have been emphasized. However, no study has analyzed the increase in the risk of depression/anxiety symptoms caused by the pay system in Korea. This study aimed to reveal the association between performance-based pay systems and symptoms of depression/anxiety, using data from the fifth Korean Working Conditions Survey. Depressive/anxiety symptoms were assessed using “yes” or “no” questions regarding medical problems related to depression/anxiety. The performance-based pay system and job stress were estimated using self-response answers. Logistic regression analyses were conducted to determine the association between performance-based pay systems, job stress, and symptoms of depression/anxiety using data from 27,793 participants. The performance-based pay system significantly increased the risk of the symptoms. Additionally, risk increments were calculated after grouping by pay system and job stress. Workers with two risk factors had the highest risk of symptoms of depression/anxiety for both sexes (male: OR 3.05; 95% CI 1.70–5.45; female: OR 2.15; 95% CI 1.32–3.50), implying synergistic effect of performance-based pay system and job stress on depression/anxiety symptoms. Based on these findings, policies should be established for early detection and protection against the risk of depression/anxiety.

## 1. Introduction

A wage system has changed extensively over the years with increasing income levels and structural changes in the industry. New technology, occupation and business organization have arisen, as economic structure has changed [1]. Subsequently, various types of wage systems have developed in accordance with the development of economy to optimize the profit model. In this process, wage system in which pay is decided by performance has emerged in several fields such as software sales, agriculture and private education [2]. In addition, the development of performance-based pay system is expected to be related to corporate social responsibility and business ethics, the importance of which is recently emphasized in company management. According to corporate social responsibility, business organization has responsibility to create company profit and share for shareholders. Thus, the management of the company may offer wage related to performance to maximize the profit. In addition, business enterprise tends to preserve good employees in accordance with business ethics. Therefore, the management may reward those who perform well to retain competent workers [3].

In Korea, a seniority-based pay system, wherein the income increased with the length of service, was a major wage system before the foreign exchange crisis in 1997 [4]. During the crisis, performance-based pay systems were implemented extensively in various forms among diverse occupations based on studies that report higher labor productivity associated with the system [5]. In 2021, 32.1% of companies paid wages by performance-based pay systems among enterprises with 100 or more employees, and an additional 17.1% of companies planned to adopt the system [6].

In a performance-based pay system, income is based on worker performance. Positive and negative effects of this system have been reported in several studies. While some have reported a 44% elevation in the productivity effects per employee after implementing a performance-based pay system [7], adverse effects of the system, such as economic, psychological, and occupational health issues, have also been revealed in various fields. Performance-pay positively correlates with musculoskeletal issues, including back pain and repetitive stress injuries, among workers in the United Kingdom [8]. Additionally, the risk of work-related injury increased among blue-collar workers in the United States after adopting a performance-based pay system [9].

Recent studies have reported an association between performance-based pay systems and mental health. Path analysis revealed that increasing adoption of performance-pay results in burnout which is a syndrome characterized by emotional and physical exhaustion among public enterprise workers in Korea [10]. Dahl and Pierce determined that the system increased the dosage rate of selective serotonin reuptake inhibitors (SSRI) and benzodiazepines by 6% in Denmark [11]. In addition, participants who are not paid by performance-based pay system reported better mental health, assessed by SF-12, compared to those who are paid by the system [12]. After the adoption of performance-based pay system, the additional income accounts for a large percentage of household spending [13]. Thus, the competition for maintaining the performance occurred, and it is expected to lead to mental health problems [10].

Emotional suppression which was used as confounder in the analyses is type of emotional regulation which inhibits uncomfortable emotional expression [14]. Generally, most of workers unavoidably suppressed their negative emotion in front of coworkers or clients. Emotional suppression is regarded as important risk factor of major depressive and anxiety disorders [15]. Thus, we adjusted for frequency of emotional suppression in the regression and interaction analysis. 

Mental health problems, including depression and anxiety, are serious social issues that can damage one’s family and decrease worker productivity, contrary to the intentions of the performance-based pay system [16]. Depression and anxiety are different constructs, but over 50% of patients who have depressive disorder report anxiety disorder in a lifetime [17]. In addition, similarities in biological pathogenesis of stress-induced anxiety and depression such as blood–brain barrier inflammation and leakage have been revealed [18]. Furthermore, no study has analyzed the risk increment for symptoms of depression/anxiety caused by the performance-based pay system in South Korea. We set out to reveal the association between the system and the reporting symptoms of depression and anxiety in paid workers in South Korea. In addition, this study aimed to assess the additive effect of the performance-based pay system and perceived job stress on the reporting symptoms of mental illness. 

## 2. Materials and Methods

### 2.1. Study Design and Study Population

This study used data from the fifth Korean Working Conditions Survey (KWCS), conducted by the Korea Occupational Safety and Health Agency in 2017. The KWCS is a triennial cross-sectional survey aimed at establishing an industrial accident prevention policy by investigating employment and labor environment. A total of 50,205 economically active workers aged >15 years participated in the fifth KWCS. We used the inclusion and exclusion criteria to extract the sample population in accordance with the study aims. First, participants who were not paid workers were excluded from this study (n = 20,097). Second, participants who did not answer one or more questions included as variables for the analysis (n = 2315) were excluded. After inclusion and exclusion, 27,793 participants were enrolled in this study.

### 2.2. Measures

#### 2.2.1. Depression/Anxiety Symptoms

The presence of reporting symptoms of depression/anxiety was assessed using the following question: “Have you ever experienced the following medical conditions in the past year?”. The possible answers to the question were either “yes” or “no.” Among the several medical problems, depression and anxiety symptoms were included in the analysis. Participants who experienced at least one depression or anxiety symptom were classified into the symptomatic group.

#### 2.2.2. Performance-Based Pay System

Participants replied to the “yes” or “no” question about the payment system as follows: “Did you get paid based on your performance after providing service at your workplace based on your work in the previous week?” Workers who responded “yes” were assigned to the performance-based pay system group.

#### 2.2.3. Perceived Job Stress

Perceived job stress was estimated by asking questions about the frequency of experiencing stress from work. The possible answers were “always”, “almost always”, ”sometimes”, “almost never”, and “never”. We regrouped the answers into two groups: always including “always” and “almost always”. sometimes/rarely including ”sometimes”,“almost never,” and “never”.

### 2.3. Covariates

This study used eight covariates: sex, age, educational level, weekly working hours, employment type, occupational classification, monthly income, and emotion suppression during working hours. We analyzed the data after stratifying the population by sex to consider sex differences in symptoms. Age was categorized into five groups: 15–29, 30–39, 40–49, 50–59, and ≥60. Educational level was classified as middle school or lower, high school, college, or higher. The grouping of weekly working hours was based on the Labor Standards Act of Korea. The limits were between 40 h and 52 h. Weekly working hours were grouped as 40 or under, 41–52, and over 52. Employment types were classified as regular, temporary, or daily. Regular workers were defined as employees with more than a year of an employment contract or without a term employment contract. Temporary workers were defined as employees with more than a month and less than a year of employment contracts. Daily workers were defined as employees with less than a month of an employment contract. The occupational classification was regrouped into office, service, and manual workers. Office workers included technicians, associate professionals, managers, and other professionals. Service workers included sales, service, and clerical support workers. Manual workers included assemblers, skilled agricultural/forestry/fishery workers, elementary workers, craft and related trade workers, and plant/machine operators. Monthly income was grouped into three categories by division into three equal parts. The limits were ₩1,800,000 on the lower side and ₩3,000,000 on the higher side. Emotion suppression was assessed based on the frequency of concealing feelings during working hours. Workers who answered “always” and “almost always” were classified as the “always” group. The “sometimes” group included workers who answered “sometimes”, and the “rarely” group included workers who answered “almost never” and “never”.

### 2.4. Statistical Analysis

A comparison of the frequency of age, educational level, weekly working hours, employment type, occupational classification, monthly income, size of the workplace, emotion suppression during working hours, job stress, and performance-based pay system was investigated after stratification by sex. We used chi-square analysis to identify differences in the rate of reporting symptoms of depression/anxiety according to demographic characteristics.

Multivariate logistic regression analysis was used to calculate odds ratios (ORs) and 95% confidence intervals (95% CIs) for reporting symptoms of depression/anxiety based on a performance-based pay system. In this analysis, we used Models A, B, and C by varying the adjustment for various factors. Model A controlled for demographic variables, including age and educational level. Model B controlled for the demographic variables in Model A and occupational variables, including employment type, occupational classification, monthly income, weekly working hours, and workplace size. Model C controlled for demographic, occupational, and job environmental variables, including emotion suppression during working hours and job stress.

An additional logistic regression model was used after grouping by performance-based pay system and job stress to analyze the additive effect of the two variables. We used relative excess risks of interaction (RERI), attributable proportion (AP), and synergistic index (SI) to analyze synergistic effect. The 95% CI of RERI and AP > 0 or the 95% CI of SI > 1 indicate that there is synergistic effect of performance-based pay system and job stress [19].

Confounders included age [20], educational level [21], weekly working hours [22], employment type [23], occupational classification [24], monthly income [25], and suppressing emotion during working hours [26]; they were determined to be significantly related to depression/anxiety symptoms, and they were controlled. Finally, variance inflation factors (VIFs) in the logistic model were calculated to confirm the presence of multicollinearity. All statistical analyses were conducted using the R software (version 4.1.3; R Foundation for Statistical Computing, Vienna, Austria).

## 3. Results

General characteristics based on adoption of performance-based pay system in this study are shown in Table 1. The adoption rate in male workers (5.8%) was lower than that in female workers (8.5%). Workers who experience performance-based pay system were more likely to report symptoms of depression/anxiety and job stress in both sexes. Furthermore, educational level, working hours, monthly income, occupational classification, and emotional suppression frequency were significantly related to the adoption of performance-based pay system. 

Table 2 shows the sociodemographic characteristics of the study population with respect to reporting symptoms of depression and anxiety. The prevalence of reporting symptoms of depression and anxiety was higher among female workers (3.8%) than in male workers (3.6%). Age and education level were significantly related to reporting symptoms of depression/anxiety in both sexes. Male workers aged between 50 and 59 had the highest prevalence of reporting symptoms of depression and anxiety (4.4%), while female workers aged ≥60 years had the highest prevalence (4.7%). In addition, male and female workers with below middle school education were more likely to report symptoms of depression or anxiety than other groups. Monthly income was significantly associated with symptoms among both male and female participants. Symptoms were more common among workers who experienced emotion suppression during their working hours.

The effects of the performance-based pay system on reporting symptoms of depression/anxiety are summarized in Table 3. We conducted a logistic regression analysis of the three models to check the robustness of this study. In Model A, we adjusted for demographic variables such as age and educational level. In Model B, demographic and occupational variables were adjusted, including employment type, occupational classification, monthly income and weekly working hours. In Model C, demographic variables, occupation variables, and psychological variables such as emotion suppression and job stress were included in the adjustment. In all models, we stratified the participants by sex, and the referent group consisted of workers not working in the performance-based pay system. Workers within the performance-based pay system had significantly higher risks than those in the reference group (OR 1.64; 95% CI 1.20–2.20 for males, OR 1.59; 95% CI 1.22–2.06 for females).

Table 4 shows the interaction effect of the performance-based pay system and job stress on reporting symptoms of depression and anxiety. In this analysis, four groups were designed by grouping the participants based on their inclusion in the performance-based pay system and job stress for each sex. We controlled for age, educational level, weekly working hours, employment time, occupational classification, monthly income, workplace size, and emotion suppression during working hours. The reference group included workers not working in a performance-based pay system and sometimes/rarely experiencing stress. The ORs for the “No performance-based pay system and always experience job stress” group were 2.18 (95% CI 1.77–2.67) for males and 1.82 (95% CI 1.49–2.22) for females. In addition, workers with two risk factors were more likely to suffer from symptoms of depression/anxiety in both males (OR 3.05; 95% CI 1.70–5.45) and females (OR 2.15; 95% CI 1.32–3.50). This result implied the presence of an additive effect of the performance-based pay system and job stress on reporting symptoms of depression/anxiety. The odds ratio for the “Performance-based pay system and sometimes/rarely experience job stress” group was not significant.

For interaction analysis in male workers, RERI was 0.79 (95% CI -0.39–2.37), indicating that the excess risk due to interaction was caused by interaction between performance-based pay system and job stress. AP was 0.26 (95% CI 0.05–0.41) and it implied that in 26% of patients exposed to two factors, it was caused by the interaction effect. SI was 1.62 (95% CI 1.11–2.36). In female workers, RERI was 0.64 (95% CI 0.20–1.48), AP was 0.30 (95% CI 0.12–0.45), and SI was 2.28 (95% CI 1.06–4.93). For both sexes, the 95% CI of SI were higher than 1.00, suggesting that there was synergistic effect of two variables. 

## 4. Discussion

This study revealed that a performance-based pay system is associated with reporting symptoms of depression/anxiety among wage workers, consistent with the findings from previous studies. Previous research has discovered that the performance-based pay system significantly elevates the anxiety level and usage of SSRI and benzodiazepines, which are the primary drugs for depressive and anxiety disorders [11,27]. However, to the best of our knowledge, the risk increment for depression/anxiety caused by a performance-based pay system has not yet been analyzed. The results of the three models using different adjustment variables were consistent with each other. Furthermore, we detected a synergistic interaction between the participants’ symptoms with a performance-based pay system and job stress. The results implied that participants who worked in performance-based pay systems and frequently experienced job stress had an even higher risk of symptoms of depression/anxiety.

The mechanism by which the performance-based pay system triggers depression/anxiety symptoms remains unclear. However, adverse effects of performance-based pay systems that could lead to the occurrence of depression/anxiety have been revealed in previous research. First, excessive and aggressive competition can occur when workers strive for better performance. In addition, in most performance-based pay systems, intra-organizational competition is essential for increasing outcomes and incentives [28]. The system could increase employee conflict and workplace bullying during the competition for incentives [29]. In addition, Glaser D. reported that a performance-based pay system elevates the harmful behavior between coworkers [13]. The competition and harmful behavior can be explained by “crowding out” effect. Additional income from the performance-based pay system may become the crucial part of pay after the adoption of the pay system. The additional pay could be critical for their spending such as expense of monthly rent and childcare and their performance may become important factor to household. Thus, competition and conflict between workers become too severe to maintain their achievements. Secondly, a performance-based pay system poses a threat of failing to earn additional income [28]. Thirdly, the system can elevate the risk of emotional exhaustion of employees, and the risk increases in systems that use high performance pay [10]. 

Discord between colleagues at work, the threat of losing incentives, and burnout could be stressors that may ultimately lead to depression symptoms [30,31]. First of all, organization where there is higher conflict between coworkers have 20.5% higher risks for worker depression compared to organization where there is no conflict [31]. The conflict could be acting as a stressor which causes symptoms of depression/anxiety. Especially the risk of current major depressive episode is significantly associated with the experience of burnout [32]. The mechanism of burnout causing symptoms of depression/anxiety is not clearly revealed. However, the similarity of occupational risk factors and biological mechanism is supposed to be related to the association [30]. In addition, performance-based pay system intrinsically represents zero-sum system and it results in intra-organization competition, bullying and emotional exhaustion. Therefore, the adverse effect of performance-based system is associated to a flaw in the design. 

As shown in Table 4, the risk for reporting symptoms of depression/anxiety increased as the number of risk factors increased among both male and female workers. The results showed an additional interaction between the performance-based pay system and job stress on symptoms. The reasoning behind the interaction of the two risk factors with depression/anxiety symptoms has not been identified. However, job stress may reinforce the predicted mechanism of the system’s effect on depression/anxiety. Chronic job stress is reportedly an important cause of burnout syndromes [33]. Furthermore, Silvia et al. revealed that job stress significantly impacts workplace conflict [34]. Burnout and workplace conflict are mechanisms by which a performance-based pay system drives individuals toward depression/anxiety. Thus, performance-based pay systems and job stress trigger symptoms depression/anxiety via a common pathway and may have a synergistic effect.

The factor which is expected to have effect on the association between performance-based pay system and the symptoms is presenteeism. Presenteeism is a phenomenon of workers continuing to work despite feeling sick. Sick leave may cause loss of performance-related incentives in the adopted performance-based pay system [35]. Therefore, the pay system could elevate the number of presenteeism. Cho et al. reported that presenteeism increased the risk of psychosocial factors such as discrimination, violence and bullying in the workplace [36]. These conditions are known as factors that negatively affect mental health symptoms [37,38]. In addition, the pay system may prevent the opportunity of treatment for mental health symptoms and directly affect the frequency and severity of depression/anxiety symptoms. 

In this study, all ORs for reporting symptoms of depression/anxiety were higher among male than female workers. According to previous research, occupational risk factors, such as job stress and income, increase the risk of depression among breadwinner workers [39]. Because the breadwinner is a household’s primary income earner, jobs are an important source of livelihood, and occupational risk factors bring heavier burdens to these individuals. Considering that breadwinners are much more common in males than in females in Korea, the difference in susceptibility to depression/anxiety symptoms between males and females is explained by the vulnerability of breadwinners. In addition, the susceptibility of risk factors for symptoms such as job stress of depression and anxiety are quite different depending on sex. Previous studies reveal the sex differences in occupational hazard exposures for same occupation workers. Male workers tend to be exposed to physical occupational hazards such as noise, heat stress and physical violence. On the other hand, female workers are more likely to experience psychological occupational hazards such as verbal abuse, bullying, job stress and burnout [40,41]. Thus, female workers are more susceptible to job stress and have more chance to experience emotional distress. The situation also affects our results for female workers, and sex differences should be considered for discussion about the intervention to resolve the problems of the pay system.

In Table 4, ORs of the pay system adoption group with low job stress were not significant among both sexes. This result can be explained by positive effect of experiencing low job stress. Low job stress may have a preventive effect on negative affectivity of workers [42]. Low negative affectivity can reduce the possibility of burnout syndrome and interpersonal conflict in workplace which are significant factors through which performance-based pay system causes depression/anxiety [43,44]. Furthermore, lowering job stress reduces vulnerability to depressive symptoms [42], For these reasons, performance-based pay system did not elevate risk for reporting symptoms of depression/anxiety in low job stress group. 

We determined that workers with both risk factors showed a much higher risk of symptoms of depression/anxiety than other workers. Thus, adopting a performance-based pay system should be sublated to occupational classes such as ambulance workers and police officers who frequently experience inevitable job stress. Additionally, employers must evaluate job stress before adopting a performance-based pay system to lower the prevalence of reporting symptoms of depression/anxiety. For companies that have already adopted the system, screening for depression and anxiety is necessary for the early detection of mental disease.

This study had some limitations. Firstly, we used data from a cross-sectional study of KWCS. Thus, we could not reveal a causal relationship between the performance-based pay system, job stress, and reporting symptoms of depression/anxiety. An additional analysis of longitudinal data is required to identify the causality of the variables. Secondly, we used single-item scale assessment tool of perceived job stress and emotional suppression which were variables for analyses. The multiple-item scale of job stress and emotional suppression can offer a more accurate assessment than the single-item scale. The third limitation was that our analysis did not consider other confounders, including time of pay system adoption, work intensity, conflict between workers and underlying disease of the participants. However, job stress could reflect work intensity of the participants and conflict between coworkers. Furthermore, we analyzed the relationship between performance-based pay system, job stress and symptoms of depression/anxiety in association level. Thus, the confounders which we did not consider had minimal effect on our results. Finally, we used questionnaire items to assess the dependent variable in this study, not the DSM-IV, which is the standard diagnostic criterion for major depressive disorder (MDD) or anxiety disorder. Thus, no validate questionnaire that evaluates disease was used and this study, using dependent variables as reported symptoms of depressed and anxious feelings. In addition, depression and anxiety can co-exist with symptoms of various mental diseases, such as bipolar disorder, schizophrenia, and adjustment disorder. Because mental disorders have characteristic pathophysiology, the pathogenesis of depression/anxiety may depend on the causative disease [45]. Thus, we did not generalize the results to mental disorders or their pathogenesis.

Nevertheless, this study had several strengths. Most importantly, this is the first study to reveal the synergistic effect of a performance-based pay system and job stress on reporting symptoms of depression/anxiety. Thus, this study provides the evidence for establishing a screening program for depression and anxiety in vulnerable workers. Furthermore, this study is based on a well-established survey which is designed to represent population of workers in South Korea. Therefore, the results in this study also can reliably explain the case of the workers in South Korea. 

## 5. Conclusions

In conclusion, the effect of the performance-based pay system and the interaction effect between the system and job stress on reporting symptoms of depression and anxiety were clearly identified. The performance-based pay system raised the risk of symptoms of depression/anxiety and created a much higher risk of the reporting symptoms in the presence of high job stress. Our findings suggest that assessing and reducing job stress and early detection of depression/anxiety symptoms among workers within the performance-based pay systems is critical to lower risks for symptoms of depression/anxiety.

## Figures and Tables

**Table 1 ijerph-20-04065-t001:** General characteristics of the study population based on adoption of performance-based pay system and demographic variables.

	Male (N = 13,362)			Female (N = 14,431)		
	Performance-Based Pay System		p (χ^2^)	Performance-Based Pay System		p (χ^2^)
	Adopted	Not Adopted		Adopted	Not Adopted	
Total	775 (5.8%)	12,587 (94.2%)		1232 (8.5%)	13,199 (91.5%)	
Reporting symptoms of depression/anxiety			<0.001			<0.001
No	723 (93.3%)	12,157 (96.6%)		1157 (93.9%)	12,730 (96.4%)	
Yes	52 (6.7%)	430 (3.4%)		75 (6.1%)	469 (3.6%)	
Age (years)			0.102			<0.001
15–29	105 (13.5%)	1773 (14.1%)		77 (6.2%)	1894 (14.3%)	
30–39	188 (24.3%)	3250 (25.8%)		177 (14.4%)	2770 (21.0%)	
40–49	183 (23.6%)	3132 (24.9%)		373 (30.3%)	3473 (26.3%)	
50–59	191 (24.6%)	2578 (20.5%)		458 (37.2%)	3125 (23.7%)	
≥60	108 (14.0%)	1854 (14.7%)		147 (11.9%)	1937 (14.7%)	
Education level			0.004			<0.001
≤Middle	401 (51.7%)	7278 (57.8%)		560 (45.5%)	6466 (49.0%)	
High	290 (37.4%)	4075 (32.4%)		563 (45.7%)	4750 (36.0%)	
≥College	84 (10.8%)	1234 (9.8%)		109 (8.8%)	1983 (15.0%)	
Working hours			<0.001			0.030
≤40	342 (44.1%)	6768 (53.8%)		825 (67.0%)	8338 (63.2%)	
41–52	237 (30.6%)	3621 (28.8%)		290 (23.5%)	3456 (26.2%)	
>52	196 (25.3%)	2198 (17.4%)		117 (9.5%)	1405 (10.6%)	
Monthly income			0.066			<0.001
<180	105 (13.5%)	2090 (16.6%)		388 (31.5%)	6362 (48.2%)	
180–299	284 (36.6%)	4311 (34.2%)		535 (43.4%)	5180 (39.2%)	
≥300	386 (49.9%)	6186 (49.2%)		309 (25.1%)	1657 (12.6%)	
Occupational classification			<0.001			<0.001
Office worker	105 (13.5%)	2089 (16.6%)		234 (19.0%)	2848 (21.6%)	
Service worker	342 (44.1%)	4678 (37.2%)		835 (67.8%)	7391 (56.0%)	
Manual worker	328 (42.3%)	5820 (46.2%)		163 (13.2%)	2960 (22.4%)	
Employment type			<0.001			0.195
Regular	581 (75.0%)	10,372 (82.4%)		882 (71.6%)	9681 (73.3%)	
Temporary/daily	194 (25.0%)	2215 (17.6%)		350 (28.4%)	3518 (26.7%)	
Emotion suppression			<0.001			<0.001
Always	398 (51.4%)	4709 (37.4%)		646 (52.4%)	5538 (42.0%)	
Sometimes	277 (35.7%)	5029 (40.0%)		410 (33.3%)	4931 (37.4%)	
Rarely	100 (12.9%)	2849 (22.6%)		176 (14.3%)	2730 (20.6%)	
Job stress			<0.001			<0.001
Always	325 (41.9%)	3800 (30.2%)		455 (36.9%)	3652 (27.7%)	
Sometimes/Rarely	450 (58.1%)	8787 (69.8%)		777 (63.1%)	9547 (62.3%)	

Values are presented as number (%).

**Table 2 ijerph-20-04065-t002:** General characteristics of the study population based on reporting symptoms of depression/anxiety and demographic variables.

	Male (N = 13,362)			Female (N = 14,431)		
	Reporting Symptoms of Depression/Anxiety		p (χ^2^)	Reporting Symptoms of Depression/Anxiety		p (χ^2^)
	Yes	No		Yes	No	
Total	482 (3.6%)	12,880 (96.4%)		544 (3.8%)	13,887 (96.2%)	
Age (years)			0.002			0.002
15–29	47 (2.5%)	1831 (97.5%)		64 (3.3%)	1971 (96.7%)	
30–39	99 (2.9%)	3339 (97.1%)		85 (2.9%)	2862 (97.1%)	
40–49	138 (4.2%)	3177 (95.8%)		139 (3.6%)	3707 (96.4%)	
50–59	122 (4.4%)	2647 (95.6%)		158 (4.4%)	3425 (95.6%)	
≥60	76 (3.9%)	1886 (96.1%)		98 (4.7%)	1986 (95.3%)	
Missing	0	0		0	0	
Education level			<0.001			<0.001
≤Middle	69 (5.2%)	1249 (94.8%)		103 (4.9%)	1989 (95.2%)	
High	148 (3.4%)	4217 (96.6%)		218 (4.1%)	5095 (95.9%)	
≥College	265 (3.5%)	7414 (96.5%)		223 (3.2%)	6803 (96.8%)	
Missing	0 (0.0%)	12(100.0%)		0 (0.0%)	11 (100.0%)	
Working hours			0.027			0.027
≤40	192 (2.7%)	6918 (97.3%)		358 (3.9%)	8805 (96.1%)	
41–52	187 (4.9%)	3671 (95.1%)		117 (3.1%)	3629 (96.9%)	
>52	103 (4.3%)	2291 (95.7%)		69 (4.5%)	1527 (95.5%)	
Missing	0	0		0	0	
Monthly income			0.001			<0.001
<180	90 (4.1%)	2195 (95.9%)		290 (4.3%)	1887 (95.7%)	
180–299	151 (3.3%)	4444 (96.7%)		175 (3.1%)	5540 (96.9%)	
≥300	241 (3.7%)	6331 (96.3%)		79 (4.0%)	6460 (96.0%)	
Missing	34 (3.5%)	942 (96.5%)		46 (4.8%)	919 (95.2%)	
Occupational classification			0.230			0.230
Office worker	82 (3.7%)	2112 (96.3%)		101 (3.3%)	2981 (96.6%)	
Service worker	169 (3.4%)	4851 (96.6%)		316 (3.8%)	7910 (96.2%)	
Manual worker	231 (3.8%)	5917 (96.2%)		127 (4.1%)	2996 (95.9%)	
Missing	0	0		0	0	
Employment type			0.211			0.211
Regular	367 (3.4%)	10,586 (96.7%)		385 (3.6%)	10,178 (96.4%)	
Temporary/daily	115 (4.8%)	2294 (95.2%)		159 (4.1%)	3709 (95.9%)	
Missing	0	0		0	0	
Emotion suppression			<0.001			<0.001
Always	251 (4.9%)	4856 (95.1%)		290 (4.7%)	5894 (95.3%)	
Sometimes	159 (3.0%)	5147 (97.0%)		188 (3.5%)	5153 (96.5%)	
Rarely	72 (2.4%)	2877 (97.6%)		66 (2.3%)	2840 (97.6%)	
Missing	0 (0.0%)	5 (100.0%)		0 (0.0%)	7 (100.0%)	
Performance-based pay system			<0.001			<0.001
Adopted	52 (6.7%)	723 (93.3%)		75 (6.1%)	1157 (93.9%)	
Not adopted	430 (3.4%)	12,157 (96.6%)		469 (3.6%)	12,730 (96.4%)	
Missing	1 (8.3%)	11 (91.7%)		0 (0.0%)	12 (100.0%)	
Job stress			<0.001			<0.001
Always	248 (6.0%)	3877 (94.0%)		243 (5.9%)	3864 (94.1%)	
Sometimes/Rarely	234 (2.5%)	9003 (97.5%)		301 (2.9%)	10,023 (97.1%)	
Missing	0 (0.0%)	5 (100.0%)		0 (0.0%)	7 (100.0%)	

Values are presented as number (%).

**Table 3 ijerph-20-04065-t003:** The association between performance-based pay system and reporting symptoms of depression/anxiety by the logistic regression model.

	Male (N = 13,362)			Female (N = 14,431)		
	Model A	Model B	Model C	Model A	Model B	Model C
Performance-based pay system						
	Reference	Reference	Reference	Reference	Reference	Reference
	2.01 (1.47–2.68)	1.90 (1.39–2.54)	1.64 (1.20–2.20)	1.73 (1.33–2.22)	1.74 (1.33–2.24)	1.59 (1.22–2.06)
Age (years)						
15–29	Reference	Reference	Reference	Reference	Reference	Reference
30–39	1.15 (0.81–1.65)	1.29 (0.89–1.88)	1.31 (0.90–1.91)	0.89 (0.64–1.24)	0.90 (0.64–1.26)	0.90 (0.64–1.26)
40–49	1.68 (1.21–2.38)	1.93 (1.35–2.80)	1.98 (1.38–2.88)	1.05 (0.78–1.43)	1.05 (0.78–1.44)	1.08 (0.79–1.47)
50–59	1.71 (1.22–2.43)	1.96 (1.36–2.87)	2.05 (1.42–3.00)	1.17 (0.86–1.61)	1.16 (0.85–1.60)	1.20 (0.87–1.66)
≥60	1.27 (0.84–1.92)	1.32 (0.86–2.02)	1.34 (0.88–2.06)	1.12 (0.75–1.68)	1.11 (0.74–1.68)	1.22 (0.81–1.85)
Education level						
≥College	Reference	Reference	Reference	Reference	Reference	Reference
High	0.61 (0.44–0.85)	0.66 (0.47–0.93)	0.64 (0.46–0.91)	0.83 (0.61–1.13)	0.81 (0.59–1.12)	0.79 (0.58–1.10)
≤Middle	0.65 (0.47–0.92)	0.79 (0.53–1.16)	0.76 (0.51–1.13)	0.70 (0.50–0.99)	0.68 (0.46–1.00)	0.68 (0.46–1.00)
Working hours						
≤40		Reference	Reference		Reference	Reference
41–52		1.99 (1.61–2.45)	1.95 (1.58–2.41)		0.83 (0.67–1.03)	0.82 (0.66–1.01)
>52		1.76 (1.36–2.26)	1.64 (1.27–2.11)		1.19 (0.90–1.55)	1.15 (0.87–1.51)
Monthly income						
≥300		Reference	Reference		Reference	Reference
180–299		0.94 (0.74–1.17)	0.97 (0.77–1.22)		0.77 (0.58–1.02)	0.79 (0.60–1.05)
<180		1.22 (1.36–2.26)	1.33 (0.95–1.83)		1.04 (0.78–1.39)	1.11 (0.83–1.49)
Occupational classification						
Office worker		Reference	Reference		Reference	Reference
Service worker		0.87 (0.67–1.15)	0.87 (0.67–1.16)		1.00 (0.78–1.29)	1.01 (0.79–1.31)
Manual worker		0.79 (0.59–1.08)	0.83 (0.61–1.13)		0.87 (0.62–1.21)	0.93 (0.67–1.31)
Employment type						
Regular		Reference	Reference		Reference	Reference
Temporary/daily		1.50 (1.16–1.94)	1.58 (1.22–2.04)		0.89 (0.72–1.11)	0.94 (0.76–1.17)
Emotion suppression						
Rarely			Reference			Reference
Sometimes			1.11 (0.84–1.48)			1.51 (1.14–2.03)
Always			1.50 (1.14–2.00)			1.74 (1.32–2.34)
Job stress						
Sometimes/Rarely			Reference			Reference
Always			2.23 (1.83–2.71)			1.95 (1.62–2.35)

Values are presented as odds ratio (95% confidence interval). Model A: Adjusted for demographic variables, including age and educational level. Model B: Model A + occupational variables, including employment type, occupational classification, monthly income and weekly working hours. Model C: Model B + psychological variables including emotion suppression during working hours and job stress.

**Table 4 ijerph-20-04065-t004:** The interaction effect of performance-based pay system and job stress on depressive/anxiety symptoms, stratified by sex.

Job Stress	Male		Female	
	Performance-Based Pay System		Performance-Based Pay System	
	Not Adopted	Adopted	Not Adopted	Adopted
Sometimes/ Rarely	1.00 (Referent)	1.09 (0.37–3.20)	1.00 (Referent)	0.69 (0.29–1.65)
Always	2.18 (1.77–2.67)	3.05 (1.70–5.45)	1.82 (1.49–2.22)	2.15 (1.32–3.50)
RERI	0.79 (−0.39–2.37)		0.64 (0.20–1.48)	
AP	0.26 (0.05–0.41)		0.30 (0.12–0.45)	
SI	1.62 (1.11–2.36)		2.28 (1.06–4.93)	

Adjusted for age, educational level, weekly working hours, employment type, occupational classification, monthly income, workplace size, and emotion suppression during working hours.

## Data Availability

Publicly available datasets were used in this research and the data can be found on the site https://oshri.kosha.or.kr/oshri/researchField/workingEnvironmentSurvey.do (accessed on 9 February 2022).

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
