# Peer review of "Performance-Based Pay System and Job Stress Related to Depression/Anxiety in Korea: Analysis of Korea Working Condition Survey"

_ijerph, 2023, doi:10.3390/ijerph20054065_

Round 1
Reviewer 1 Report
1. Please clarify whether the impact of the performance-based pay system of concern to this study is a flaw in the design of the system or a problem in the process of implementing the system.
2. Job stress at work is not always a bad thing. How can you ensure that the job stress that is of concern to this study will necessarily have a negative impact on job performance?
3. Why did this study construct three models using different covariates? Why not just construct one?
4. Since previous studies have found the risk of employees’ emotional exhaustion, what additional theoretical contribution does the current study make by extending this negative effect to depression or anxiety? Are these variables fundamentally different?
5. Please clarify that the confounding factors not considered in this study are not fatal to the results. Otherwise, this could be a limitation that overturns the conclusions of the entire article.
6. Please add as much as possible to the mechanisms behind the impact of the performance-based pay system. In other words, the authors should reflect their valuable insights as academic experts, not just complete a data analysis that can be achieved by laymen.
Author Response
We also attach the response letter in Word file.
Reviewer #1:
We deeply thank you for taking the time to read our manuscript and to provide feedback. Your comments were highly insightful and helped us to improve the quality of our study. We addressed the concerns point by point in the following pages.
Point P1
Please clarify whether the impact of the performance-based pay system of concern to this study is a flaw in the design of the system or a problem in the process of implementing the system.
(Response)
We are really grateful for reviewer’s comment. We aimed to analyze the flaw and intrinsic problems in the design of the system. The risk elevation of the performance-based pay system is significant after the adjustment of several occupational variables. Also, according to the previous research [1] [2], performance-based pay system intrinsically represents zero-sum system and it makes intra-organizational bullying and deviance. Also, the bullying and deviance between workers are the important factor for symptoms of depression/anxiety. Therefore, impact of the performance-based pay system of concern to the study is a flaw in the intrinsic design of the pay system. We added the sentence and the sentence is showed in the following paragraph.
[1] Samnani, A.-K.; Singh, P. Performance-enhancing compensation practices and employee productivity: The role of workplace bullying.
[2] Gläser, D.; van Gils, S.; Van Quaquebeke, N. Pay-for-performance and interpersonal deviance: Competitiveness as the match that lights the fire.
Line 288-291: “Also, performance-based pay system intrinsically represent zero-sum system and it makes intra-organization competition, bullying and emotional exhaustion. Therefore, the adverse effect of performance-based system is associated to a flaw in the design.”
Point P2
Job stress at work is not always a bad thing. How can you ensure that the job stress that is of concern to this study will necessarily have a negative impact on job performance?
(Response)
We deeply appreciate the reviewer’s suggestion. We also agree that job stress sometimes can be a motivation factor and stimulant for great job performance. According to various studies, job stress is important risk factors for mental illnesses including depression and mental illness can be the main cause of leave of absence and lower the job performance. Furthermore, job stress has synergistic effect with performance-based pay system to elevate risk of symptom of depression/anxiety in this study. Thus, job stress has pros and cons for job performance and we reveal one adverse effect of job stress that increase the risk of symptom of depression/anxiety.
Point P3
Why did this study construct three models using different covariates? Why not just construct one?
(Response)
We are really thankful for the reviewer’s comment on models of the regression analysis. We construct three logistic regression models in the analysis of association between performance-based pay system and symptoms of depression/anxiety. At first, the analysis of three models were performed to check the robustness of results in the study. Models all have different covariates and the risk elevation in model A, B, C were all significant. Therefore, we conclude that the robustness was verified. Also, we can analyze various odds ratios of symptoms of depression/anxiety after adjusting different types of covariates, including demographic/occupational/psychological variables. The revised paragraph is showed in the following paragraph.
Line 204-206: “The effects of the performance-based pay system on reporting symptoms of depression/anxiety are summarized in Table 3. We conducted a logistic regression analysis of the three models to check the robustness of this study.”
Editor Point P4
Since previous studies have found the risk of employees’ emotional exhaustion, what additional theoretical contribution does the current study make by extending this negative effect to depression or anxiety? Are these variables fundamentally different?
(Response)
We really appreciate for the reviewer’s comment about the description of the emotional exhaustion and burnout. Emotional exhaustion is a state of reduced accomplishment and personal identity caused by severe and prolonged physical and mental stress. Also, emotional exhaustion is important component of burnout syndrome. Burnout syndrome is a chronic state of lethargy and fatigue caused by extreme physical and mental stress.
The mechanism that burnout cause symptoms of depressive/anxiety is not clearly revealed. According to previous meta-analysis, association between burnout and depression, burnout and anxiety are significant [1]. Also, risk of current major depressive episode is significantly associated with the experience of burnout. [2] Depression/anxiety and burnout have important occupational risk factor which is job stress and share a common biologic mechanism such as DNA methylation. [1] The similarity between burnout and depression/anxiety is supposed to be related to the association. The corrected paragraph is showed in the following paragraph.
[1] Koutsimani, P.; Montgomery, A.; Georganta, K. The relationship between burnout, depression, and anxiety: A systematic review and meta-analysis
[2] Ahola, K., Honkonen, T., Isometsä, E., Kalimo, R., Nykyri, E., Aromaa, A., & Lönnqvist, J. (2005). The relationship between job-related burnout and depressive disorders—results from the Finnish Health 2000 Study
Line 284-288: “Especially, risk of current major depressive episode is significantly associated with the experience of burnout [32]. The mechanism that burnout cause symptom of depressive/anxiety is not clearly revealed. However, the similarity of occupational risk factors and biological mechanism is supposed to be related to the association [30].”
Editor Point P5
Please clarify that the confounding factors not considered in this study are not fatal to the results. Otherwise, this could be a limitation that overturns the conclusions of the entire article.
(Response)
We are grateful for reviewer’s comment on the confounders that are not considered in this study. First, the confounding factors not considered in this study are the time of pay system adoption, underlying disease of the participants, work intensity and conflict between workers. These variables were not included in the questionnaire of the survey. However, job stress could reflect work intensity of the participants and conflict between workers. Also, we analyze the association between performance-based pay system, job stress and symptoms of depression/anxiety, not a causal relationship. Thus, the confounders which we considered in this study were sufficient to derive a conclusion about the association. We revised the conclusion and revised paragraph is showed in the following paragraph.
Line 354-361: “The third limitation was that our analysis did not consider other confounders, including time of pay system adoption, work intensity, conflict between workers and underlying disease of the participants. However, job stress could reflect work intensity of the participants and conflict between coworkers. Furthermore, we analyzed the relationship between performance-based pay system, job stress and symptoms of depression/anxiety in association level. Thus, the confounders which we did not consider had minimal effect to our results.”
Editor Point P6
Please add as much as possible to the mechanisms behind the impact of the performance-based pay system. In other words, the authors should reflect their valuable insights as academic experts, not just complete a data analysis that can be achieved by laymen.
(Response)
We deeply appreciate for reviewer’s comment on the mechanisms about the impact of the performance-based pay system. We find additional theories and mechanisms about the impact of performance-based pay system. Firstly, “crowding out” effect explains how performance-based pay system causes the competition and harmful behavior. Additional income from the pay system may become the crucial part of worker’s income after the performance-based pay system was adopted. Then, the additional income could be critical for household spending and the performance of workers become important factor for their lives. Finally, competition and conflict to raise the performance may become harsh to maintain their income. Secondly, organization where there is higher conflict between workers have 20.5% higher risks for depression compared to organization where there is no conflict. The interpersonal conflict may act as a stressor which triggers depressive/anxious symptoms. The added manuscript is showed in the following paragraph.
Line 269-276: “In addition, Glaser D. found that a performance-based pay system elevates the harmful behavior between coworkers [13]. The competition and harmful behavior can be explained by “crowding out” effect. Additional income from the performance-based pay system may become the crucial part of pay after the adoption of the pay system. The additional pay could be critical for their spending such as expense of monthly rent and childcare and their performance may become important factor to household. Thus, competition and conflict between workers become severe to maintain their achievements.”
Line 281-284: “First of all, organization where there is higher conflict between coworkers have 20.5% higher risks for depression as compare to organization where there is no conflict [31]. The conflict could be acting as a stressor which causes symptoms of depression/anxiety.”

Reviewer 2 Report
Thank you for the opportunity to review this interesting manuscript. The authors analyzed the results of the Korea Working Condition Survey, to investigate the association between performance-based pay systems and depression/anxiety.
The study is of relevance due to the current debates about the pros and cons of adopting working performance-based pay systems. The manuscript is generally well-written and the methodology is sound, I have, however, a couple of points which I encourage the authors to address to improve the overall quality.
Major points:
- The results of the main variables (i.e., the prevalence of depression/anxiety, job stress, and emotional exhaustion) in the two groups are not reported and are “hidden” in the regression analyses. I strongly suggest adding a table reporting these data.
- Depression and anxiety are different constructs, with different antecedents and consequences. The authors should provide reasons for considering a single depression/anxiety variable.
- The Introduction section should be expanded to better explain the background of the study (e.g., more about the relationship between performance pay systems and mental health, what is emotion suppression and why is it important for the study).
- Perceived job stress and Emotion suppression have been assessed with single items, while it is known that multi-item scales perform better than single-item measures. Please add the use of single items as another limitation of the study.
- The authors write in the Conclusions that “The performance-based pay system raised the risk of depression/anxiety and created a much higher risk of job stress”. However, it seems to me that the analyses do not show that the performance-based pay system increases the risk of job stress. According to my understanding of Table 3, however, it seems clear that job stress is THE critical risk factor for depressive/anxiety symptoms: when job stress is low, adopting a performance-based pay system doesn’t increase the risk of mental symptoms (and it actually lowers it for female workers). Therefore, I believe that the correct takeaway message of this study is that assessing and reducing work-related stress is critical for improving workers’ mental health, and this is especially true in organizations that adopt performance-based pay systems.
Minor points:
- Page 2, row 54 “emotional exhaustion, which is called burnout, …”. This is not 100% correct, emotional exhaustion is considered to be a core component of burnout, not a synonym of burnout.
- There are some typos (e.g., page 3 rows 97-98, “always” repeated multiple times; page 4 rows 154-155 “depression and anxiety was higher among male workers (3.6%) than in female workers (3.8%).”; page 6 row 198 “relative excess risk was existed by interaction…”; page 6 row 200 “26% of mental symptoms in patients who exposed to two factors…”), so I suggest another round of in-depth proofreading.
Author Response
We also attach the response letter in Word file.
Reviewer #2:
We deeply appreciate the reviewer for taking the time to read our manuscript and to provide feedback. Your comments were highly insightful and helped us to improve the quality of our study. In the following pages, we addressed the concerns point by point.
Point P1
The results of the main variables (i.e., the prevalence of depression/anxiety, job stress, and emotional exhaustion) in the two groups are not reported and are “hidden” in the regression analyses. I strongly suggest adding a table reporting these data.
(Response)
We really thank the reviewer’s point of view on the main variables. Firstly, we add a table about the results of main variables in the regression analyses. It includes odds ratios in the logistic regression analysis for the variables, including emotional suppression, job stress, age, education level, working hours, monthly incomes, occupational classification and employment type. Also, we added the prevalence of the symptoms for pay system adopted group and not-adopted group in the Table 1, respectively and the prevalence for general population is presented in Table 2. Secondly, we add a table that express the percentage of job stress, emotional exhaustion in Table 1 and it shows general characteristics of the study population based on adoption of performance-based pay system. The added tables are showed in the attachment file.
Point P2
Depression and anxiety are different constructs, with different antecedents and consequences. The authors should provide reasons for considering a single depression/anxiety variable.
(Response)
We are really grateful for reviewer’s comment on depression and anxiety. Firstly, we select symptoms of depression and anxiety because depressive symptom and anxiety symptom are common mental health symptom which has high prevalence rates (6.2%-11.5% for depressive symptom, 5%-14.4% for anxiety symptom). [1] Also, over 50% of patients who have depressive disorder report anxiety disorder in a lifetime. [2] Recently, stress-induced depression symptom and anxiety symptom may share biological mechanism such as blood-brain barrier inflammation and leakage [3]. We agree that depression and anxiety are different constructs and have antecedents and consequences. However, comorbidity rate is high and the similarity of biological mechanism has been revealed. Therefore, we decide to evaluate the risk elevation of symptoms of depression/anxiety, caused by performance-based pay system and job stress. We added the sentence and the sentence is showed in the following paragraph.
[1] Schafer, K. M., Lieberman, A., Sever, A. C., & Joiner, T. (2022). Prevalence rates of anxiety, depressive, and eating pathology symptoms between the pre-and peri-COVID-19 eras: A meta-analysis
[2] Kaufman, J., & Charney, D. (2000). Comorbidity of mood and anxiety disorders
[3] Welcome, M. O. (2020). Cellular mechanisms and molecular signaling pathways in stress-induced anxiety, depression, and blood–brain barrier inflammation and leakage
Line 82-86: “Depression and anxiety are different constructs, but over 50% of patients who have depressive disorder report anxiety disorder in a lifetime [17]. Also, similarities of biological pathogenesis of stress-induced anxiety and depression such as such as blood-brain barrier inflammation and leakage have been revealed [18].”
Point P3
The Introduction section should be expanded to better explain the background of the study (e.g., more about the relationship between performance pay systems and mental health, what is emotion suppression and why is it important for the study).
(Response)
We really thank the reviewer’s comment on the background of the study. Firstly, we added additional information about the relationship between performance-based pay system and mental health according to previous research. Participants which are not paid by performance-based pay system report better mental health, assessed by SF-12, compared to those which are paid by the pay system. Also, we added the theory how the pay system can cause symptoms of depression/anxiety. After the adoption of performance-based pay system, the additional income account for a large percentage of household spending. Thus, the competition for maintaining the performance occurs and the competition and conflict may act as a stressor which causes the symptoms. Furthermore, we added the information about emotional suppression and the importance of emotional suppression in our study. Emotional suppression is type of emotional regulation which inhibit uncomfortable emotion expression. Also, it is confounder in our study and we adjusted for emotional suppression frequency. We added the sentence and the sentence is showed in the following paragraph.
Line 68-70: “Also, participants who are not paid by performance-based pay system reported better mental health, assessed by SF-12, compared to those who are paid by the pay system [12].”
Line 70-73: “After the adoption of performance-based pay system, the additional income account for a large percentage of household spending [13]. Thus, the competition for maintaining the performance occur and is expected to lead to mental problems [10].”
Line 74-79: “Emotional suppression which was used as confounder in the analyses is type of emotional regulation which inhibits uncomfortable emotional expression [14]. Generally, most of workers unavoidably suppressed their negative emotion to coworkers or clients. Emotional suppression is regarded as important risk factor of major depressive disorder and anxiety disorder [15]. Thus, we adjusted for frequency of emotional suppression in the regression and interaction analysis.”
Point P4
Perceived job stress and Emotion suppression have been assessed with single items, while it is known that multi-item scales perform better than single-item measures. Please add the use of single items as another limitation of the study.
(Response)
We are really grateful for reviewer’s comment on the assessment of perceived job stress and emotional suppression. We also agree with the multiple-item scale assessment tool performs better than the single-item scale assessment and it is limitation of our study definitely. We added the limitation about the single-item scale and the sentence is showed in the following paragraph.
Line 351-354: “Secondly, we used single-item scale assessment tool of perceived job stress and emotional suppression which were variables for analyses. The multiple-item scale of job stress and emotional suppression can assess more accurately than the single-item scale.”
Point P5
The authors write in the Conclusions that “The performance-based pay system raised the risk of depression/anxiety and created a much higher risk of job stress”. However, it seems to me that the analyses do not show that the performance-based pay system increases the risk of job stress. According to my understanding of Table 3, however, it seems clear that job stress is THE critical risk factor for depressive/anxiety symptoms: when job stress is low, adopting a performance-based pay system doesn’t increase the risk of mental symptoms (and it actually lowers it for female workers). Therefore, I believe that the correct takeaway message of this study is that assessing and reducing work-related stress is critical for improving workers’ mental health, and this is especially true in organizations that adopt performance-based pay systems.
(Response)
We are really grateful for reviewer’s comment. We agree with the review’s point and the performance-based pay increased risk of depression/anxiety symptoms and correct the sentence “The performance-based pay system raised the risk of depression/anxiety and created a much higher risk of job stress” to “The performance-based pay system raised the risk of symptoms of depression/anxiety and created a much higher risk of the reporting symptoms in the presence of high job stress”. Furthermore, we added the comment that reducing work-related stress is crucial to lower prevalence rate of symptoms of depression/anxiety and revised the conclusion. We correct the sentence and the sentence is showed in the following paragraph.
Line 380-385: “The performance-based pay system raised the risk of symptoms of depression/anxiety and created a much higher risk of the reporting symptoms in the presence of high job stress. Our findings suggest that assessing and reducing job stress and early detection of depression/anxiety symptoms among workers within the performance-based pay systems is critical to lower risks for symptoms of depression/anxiety.”
Point P6
Page 2, row 54 “emotional exhaustion, which is called burnout, …”. This is not 100% correct, emotional exhaustion is considered to be a core component of burnout, not a synonym of burnout.
(Response)
We deeply appreciate the reviewer’s suggestion on the difference of emotional exhaustion and burnout. We also agree with the reviewer’s points about burnout and emotional exhaustion. Emotional exhaustion is one of the symptoms of burnout and emotional exhaustion and burnout are not a same word. We revise the sentence about the burnout and emotional exhaustion and the sentence is showed in the following paragraph.
Line 64-66: “Path analysis revealed that increasing adoption of performance-pay results in burnout which is syndrome characterized by emotional and physical exhaustion among public enterprise workers in Korea [10].”
Point P7
There are some typos (e.g., page 3 rows 97-98, “always” repeated multiple times; page 4 rows 154-155 “depression and anxiety was higher among male workers (3.6%) than in female workers (3.8%).”; page 6 row 198 “relative excess risk was existed by interaction…”; page 6 row 200 “26% of mental symptoms in patients who exposed to two factors…”), so I suggest another round of in-depth proofreading.
(Response)
We apologize for typos in our manuscript. We correct all of the typos in the reviews and review closely and proofread all of the manuscript. The corrected paragraphs were showed in the following paragraph.
Line 120-122: “We regrouped the answers into two groups: always including “always” and “almost always,” sometimes/rarely including ”sometimes,” “almost never,” and “never.””
Line 186-188: “The prevalence of reporting symptoms of depression and anxiety was higher among female workers (3.8%) than in male workers (3.6%).”
Line 238-240: “For interaction analysis in male workers, RERI was 0.79 (95% CI -0.39–2.37), indicating that the excess risk due to interaction was caused by interaction between performance-based pay system and job stress.”
Line 240-241: “AP was 0.26 (95% CI 0.05–0.41) and it implied that 26% of patients who exposed to two factors were caused by interaction effect.”

Reviewer 3 Report
Comment for the authors:
The authors are presenting an original article concerning performance-based pay systems and their correlation with stress, depression and anxiety in Korean workers. The topic is highly interesting and original. The primary outcome is the analysis of depression and anxiety in the working population, the tool used to evaluate this outcome is not validated. For this reason the work must be rejected. If the authors have the opportunity to validate the tool or demonstrate that the questionnaire used has been validated for this purpose, the following reviews are required:
1. Introduction:
The introduction (lines 32-66) starts with a sentence on wage systems that is very generic and may need to be reformulated to be easier to understand, and then shifts to the Korean focus immediately. In my opinion, a more in-depth explanation of what the authors mean may be in order, especially to frame it into economic development context (just as an example, see: Sbardella et al. Economic development and wage inequality: A complex system analysis).
Furthermore, it would be interesting to frame how this emerging model of performance-based pay relates to business ethics, corporate social responsibility and retaining good employee through an ethical workplace climate (just as an example, see: Adda et al. Business Ethics And Corporate Social Responsibility For Business Success And Growth).
The last sentence in the introduction (lines 65-66) should not be placed here. Instead of a sentence focusing on the study’s results, this sentence should be reformulated as part of the study aim.
2. Methods:
Methods section (lines 69-151) thoroughly explains the study design. Concerning the measurements, the authors explain how they evaluated the presence of anxious/depressive symptoms (lines 81-86); however, multiple times thorough the manuscript the authors then refer to participants as anxious or depressed. As they only evaluated the presence of symptoms rather than using a validated questionnaire to investigate the presence of anxiety and depression, this should be amended, changing the terms “anxiety/anxious” or “depression/depressed” with “reporting symptoms of anxiety” or “reporting symptoms of depression”.
When investigating perceived job stress (lines 94-98), the authors only capitalized one of the answers (“Sometimes”) and not the other, I think this might be an oversight. Same with emotion suppression answers (line 122).
Statistical analysis (lines 124-151) is well explained and appropriate for the study design.
3. Results:
Results are well organized, and the use of tables makes them easy to understand to the reader.
For the reasons stated above: (line 159) the authors say “suffer from depression or anxiety”, it should be “reported symptoms of depression or anxiety”.
4. Discussion:
The discussion (lines 209-286) is very interesting and touches on all the results emerged from the study.
However, gender differences in how performance-based pay relates to reported anxious/depressive symptoms, which is one of the most interesting results, is only briefly touched upon (lines 247-254). Authors found that males are more subjected to anxiety and depression in this study; however, women are generally more susceptible to work-related stress (just as an example, see: Santoro et al. Occupational hazards and gender differences: a narrative review). The authors attribute this to males being the breadwinners, but this should further discussed in my opinion.
While the discussion frames very well the mental health issues investigated, I think a paragraph could be added to contextualize the study in the Korean occupational world: how does presenteeism factor into the results? (just as an example, see: Cho et al. The association between Korean workers’ presenteeism and psychosocial factors within workplaces).
In the study limitations (lines 270-282) the authors mention that they utilize items from their questionnaire to investigate anxiety and depression; I think it should be added clearly that no validated questionnaire was used, but participants reported if they had anxious or depressed feelings. Furthermore, the authors may want to mention that during the COVID-19 pandemic years, mental health problems have been rising in the general population, and this may act as a confounding factor for this research (just as an example, see: Gualano et al. TElewoRk-RelAted Stress (TERRA), Psychological and Physical Strain of Working From Home During the COVID-19 Pandemic: A Systematic Review).
In the study strengths (lines 283-286), the authors begin what should be a list (“first”, line 283), but then only mention one strength. More should be added or the sentence should be amended.
5. Conclusions
Conclusions (lines 288-293) are consistent with results and well summarized.
Author Response
We also attach the response letter in Word file.
Reviewer #3:
We really appreciate you for taking the time to read our manuscript and to provide feedback. Your comments were highly insightful and helped us to improve the quality of our study. We agree with your comments and correct our manuscript faithfully. We addressed the concerns point by point in the following pages.
Editor Point P1
The introduction (lines 32-66) starts with a sentence on wage systems that is very generic and may need to be reformulated to be easier to understand, and then shifts to the Korean focus immediately. In my opinion, a more in-depth explanation of what the authors mean may be in order, especially to frame it into economic development context (just as an example, see: Sbardella et al. Economic development and wage inequality: A complex system analysis).
(Response)
We are grateful for the reviewer’s comment on first paragraph of introduction. We also agree with the point of view on the introduction. We correct the generic sentence and add more explanation about the change of wage system. Also, we read the article : Sbardella et al. Economic development and wage inequality: A complex system analysis and add the introduction about the association of economic development and wage system and add the reference. New occupation and technology have arisen, as economic structure has changed and various type of wage system has developed in accordance with the development of economy to optimize the profit model. The revised manuscript is showed in the following paragraph.
Line 33-39: “A wage system has changed extensively over the years with increasing income levels and structural changes in the industry. New technology, occupation and business organization have arisen, as economic structure has changed [1]. Subsequently, various type of wage system has developed in accordance with the development of economy to optimize the profit model. In this process, wage system that pay is decided by performance is emerged in several fields such as software sales, agriculture and private education [2].”
Editor Point P2
Furthermore, it would be interesting to frame how this emerging model of performance-based pay relates to business ethics, corporate social responsibility and retaining good employee through an ethical workplace climate (just as an example, see: Adda et al. Business Ethics And Corporate Social Responsibility For Business Success And Growth).
(Response)
We are thankful for review’s comment on the business ethics and corporate social responsibility. We also agree that the development of performance-based pay system is related to the business ethics and corporate social responsibility. Also, we read the article : Adda et al. Business Ethics And Corporate Social Responsibility For Business Success And Growth and added the introduction. Company preserve good employees in accordance with business ethics and the management may reward for workers who performs well to retain competent workers. The revised manuscript is showed in the following paragraph.
Line 39-46: “Also, development of performance-based pay system is expected to be related to corporate social responsibility and business ethics which the importance is emphasized in company management recently. According to corporate social responsibility, business organization has responsibility to create company profit and share for shareholders. Thus, the management of the company may give wage which is related to performance to maximize the profit. In addition, business enterprise tends to preserve good employees in accordance with business ethics. Therefore, the management may reward for workers who perform well to retain competent workers [3].”
Editor Point P3
The last sentence in the introduction (lines 65-66) should not be placed here. Instead of a sentence focusing on the study’s results, this sentence should be reformulated as part of the study aim.
(Response)
We really thank the reviewer’s point of view on last sentence of the introduction. We correct the sentence and reformulated as part of the study aim which is to assess the association between performance-based pay system, job stress and the symptoms of depression/anxiety. The revised manuscript is showed in the following paragraph.
Line 87-91: “We set out to reveal the association between the system and the reporting symptoms of depression and anxiety in paid workers in South Korea. Also, this study aimed to assess the additive effect of the performance-based pay system and perceived job stress on the reporting symptoms of mental illness.”
Editor Point P4
Methods section (lines 69-151) thoroughly explains the study design. Concerning the measurements, the authors explain how they evaluated the presence of anxious/depressive symptoms (lines 81-86); however, multiple times thorough the manuscript the authors then refer to participants as anxious or depressed. As they only evaluated the presence of symptoms rather than using a validated questionnaire to investigate the presence of anxiety and depression, this should be amended, changing the terms “anxiety/anxious” or “depression/depressed” with “reporting symptoms of anxiety” or “reporting symptoms of depression”.
(Response)
We appreciate the reviewer’s suggestion on our dependent variables. We entirely agree with the comments that questionnaire in the survey evaluate the reporting symptoms of depression and reporting symptoms of anxiety. We review all of the manuscript and correct “depression” and “anxiety” to “reporting symptoms of depression” and “reporting symptoms of anxiety”. The revised manuscript was showed in the following paragraph.
Line 106-108 “The presence of reporting symptoms of depressive/anxiety was assessed using the following question: “Have you ever experienced the following medical conditions in the past year?”.”
Line 152-154 “We used chi-square analysis to identify differences in the rate of reporting symptoms of depression/anxiety according to demographic characteristics.”
Line 155-157 “Multivariate logistic regression analysis was used to calculate odds ratios (ORs) and 95% confidence intervals (95% CIs) for reporting symptoms of depression/anxiety based on a performance-based pay system.”
Line 223-224 “Table 4 shows the interaction effect of the performance-based pay system and job stress on reporting symptoms of depression and anxiety.”
Editor Point P5
When investigating perceived job stress (lines 94-98), the authors only capitalized one of the answers (“Sometimes”) and not the other, I think this might be an oversight. Same with emotion suppression answers (line 122).
(Response)
We thank the reviewer’s comment and apologize for our mistakes. “Sometimes” should be corrected to “sometimes”. We review all of the manuscript and revise the words. The revised manuscript is showed in the following paragraph.
Line 119-122: “The possible answers were “always,” “almost always,” ” sometimes,” “almost never,” and “never.” We regrouped the answers into two groups: always including “always” and “almost always,” sometimes/rarely including ”sometimes,” “almost never,” and “never.””
Line 145-147: “The sometimes group included workers who answered “sometimes,” and the rarely group included workers who answered “almost never” and “never.””
Editor Point P6
For the reasons stated above: (line 159) the authors say “suffer from depression or anxiety”, it should be “reported symptoms of depression or anxiety”.
(Response)
We really appreciate the reviewer’s comment. We correct the “suffer from depression or anxiety” to “have reported symptoms of depression or anxiety”. The revised sentence is showed in the following paragraph.
Line 191-193: “In addition, male and female workers with below middle school education were more likely to have reported symptoms of depression or anxiety than other groups.”
Editor Point P7
However, gender differences in how performance-based pay relates to reported anxious/depressive symptoms, which is one of the most interesting results, is only briefly touched upon (lines 247-254). Authors found that males are more subjected to anxiety and depression in this study; however, women are generally more susceptible to work-related stress (just as an example, see: Santoro et al. Occupational hazards and gender differences: a narrative review). The authors attribute this to males being the breadwinners, but this should further discussed in my opinion.
(Response)
We appreciate the reviewer’s suggestion on sex and gender differences. We also agree that the discussion about sex and gender differences should be added in this manuscript. We read the article : Santoro et al. Occupational hazards and gender differences: a narrative review and other articles and added the reference. Male workers tend to be exposed physical occupational hazard such as noise, heat stress and physical violence. On the other hands, female workers are more likely to be exposed psychological occupational hazards such as verbal abuse, bullying, job stress and burnout. Therefore, we thought that female workers were more susceptible to job stress. Also, they have more chances to experience emotional distress. The added manuscript is showed in the following paragraph.
Line 321-331: “In addition, the susceptibility of risk factors for symptoms such as job stress of depression and anxiety are quite different depending on sexes. Previous studies reveal the sex differences in occupational hazard exposures for same occupation workers. Male workers tend to be exposed physical occupational hazard such as noise, heat stress and physical violence. On the other hands, female workers are more likely to experience psychological occupational hazards such as verbal abuse, bullying and job stress [40,41]. Thus, female workers were more susceptible to job stress and have more chance to experience emotional distress. The situation also affects our results for female workers and sex differences should be considered to discuss about the intervention to resolve the problems of the pay system.”
Editor Point P8
While the discussion frames very well the mental health issues investigated, I think a paragraph could be added to contextualize the study in the Korean occupational world: how does presenteeism factor into the results? (just as an example, see: Cho et al. The association between Korean workers’ presenteeism and psychosocial factors within workplaces).
(Response)
We really thank the reviewer’s comment on the presenteeism factor. We agree and are interested in the effect of presenteeism on our results. Presenteeism is defined as a state that workers continue to work despite feeling sick. Performance-based pay system can induce situation that workers cannot take sick leave to raise the performance. Presenteeism has direct effect on symptoms of depression/anxiety and causes workplace violence, bullying and discrimination which are risk factors for the symptoms. The added paragraph is showed in the following paragraph.
Line 304-313: “The factor which is expected to have effect on the association between performance-based pay system and the symptoms was presenteeism. Presenteeism is a state that workers continue to work despite feeling sick. Sick leave may cause loss of performance-related incentives while the performance-based pay system was adopted [35]. Therefore, the pay system could elevate the number of presenteeism. Cho et al reported that presenteeism increased the risk of psychosocial factors such as discrimination, violence and bullying in the workplace [36]. These conditions were known as factors that negatively affect mental health symptoms [37,38]. Also, the pay system may block opportunity to treat for mental health symptoms and directly effect on frequency and severity of depression/anxiety symptoms.”
Editor Point P9
In the study limitations (lines 270-282) the authors mention that they utilize items from their questionnaire to investigate anxiety and depression; I think it should be added clearly that no validated questionnaire was used, but participants reported if they had anxious or depressed feelings. Furthermore, the authors may want to mention that during the COVID-19 pandemic years, mental health problems have been rising in the general population, and this may act as a confounding factor for this research (just as an example, see: Gualano et al. TElewoRk-RelAted Stress (TERRA), Psychological and Physical Strain of Working From Home During the COVID-19 Pandemic: A Systematic Review).
(Response)
We appreciate the reviewer’s point on limitation. We also agree that the questionnaire in this study evaluates the reporting symptoms of depression and anxiety. Following the reviewer’s comment, we added the limitation about questionnaire in the survey. The added paragraph is showed in the following paragraph.
Also, we discuss about the limitation that COVID-19 pandemic acts as a confounding factor for this study. The study used the fifth Korea Working Condition Survey which was conducted in 2017 and the outbreak of COVID-19 pandemic occurred in December 2019. Therefore, we think that COVID-19 pandemic has minimal effect on our results.
Line 361-365: “Finally, we used questionnaire items to assess the dependent variable in this study, not the DSM-IV, which is the standard diagnostic criteria for major depressive disorder (MDD) or anxiety disorder. Thus, no validate questionnaire that evaluate disease was used and this study used dependent variables as reported symptoms of depressed and anxious feelings.”
Editor Point P10
In the study strengths (lines 283-286), the authors begin what should be a list (“first”, line 283), but then only mention one strength. More should be added or the sentence should be amended.
(Response)
We thank for the reviewer’s suggestion on the strengths. We added the more strength of our study which is that the study is based on a well-established and population-based survey. The added paragraph is showed in the following paragraph.
Line 373-376: “Furthermore, this study is based on a well-established survey which is designed to represent population of workers in South Korea. Therefore, the results in this study also can reliably explain the situation of the workers in South Korea.”

Round 2
Reviewer 2 Report
In this revised version of their manuscript, the authors have correctly addressed all the concerns reported in my previous review. Therefore, I believe it could now be accepted for publication in IJERPH.
Reviewer 3 Report
The work is very well written and the statistical analyses are valid. It remains a strong weakness not to have used validated questionnaires for the analysis of anxiety and depression.